# Influence of light availability and soil productivity on insect herbivory on bilberry (*Vaccinium myrtillus* L.) leaves following mammalian herbivory

**Marcel Schrijvers-Gonlag***, **Christina Skarpe, Harry Peter Andreassen†**

Campus Evenstad, Faculty of Applied Ecology, Agricultural Sciences and Biotechnology, Inland Norway University of Applied Sciences, Koppang, Norway

† Deceased.
* marcel.schrijversgonlag@inn.no

**Data Availability Statement:** The dataset and script used for the presented analyses are stored in

## Abstract

Vegetative parts of bilberry (*Vaccinium myrtillus*) are important forage for many boreal forest mammal, bird and insect species. Plant palatability to insects is affected by concentration of nutrients and defense compounds in plants. We expected that palatability of bilberry leaves to insect herbivores is influenced by light availability and soil productivity (both affecting nitrogen concentration and constitutive carbon-based defense compound concentration) and herbivory by mammals (affecting nitrogen concentration and induced carbon-based defense compound concentration). We studied bilberry leaf herbivory under different light availability, soil productivity and mammalian herbivory pressure in small sampling units (1m x 1m) in boreal forest in Norway. We used generalized linear mixed models and generalized additive mixed models to model insect herbivory on bilberry leaves as a function of shade, soil productivity and mammalian herbivory. Observed insect herbivory on bilberry leaves increased with increasing shade levels. Predicted insect herbivory increased with increasing previous mammalian herbivory at high shade levels and this response was magnified at higher soil productivity levels. At low to intermediate shade levels, this response was only present under high soil productivity levels. Our results indicate that light availability is more important for variation in bilberry leaf palatability than soil nutrient conditions.

## Introduction

Bilberry (*Vaccinium myrtillus* L.) is a deciduous clonal dwarf shrub with evergreen shoots that is abundant on many nutrient-poor soils in the boreal forest region of Scandinavia [1–6]. The vegetative parts of bilberry are important forage for many mammal, bird and insect species [7–13]. Insect herbivores can be indirectly affected by mammalian herbivores, which can modify food quantity, e.g., plant cover and biomass, and food quality, e.g., nutrient concentration [14–16] and the concentration and composition of chemical defense metabolites in plants [17–19].

the DataverseNO database and available at https://doi.org/10.18710/89MLBP.

**Funding:** This study is a part of the BEcoDyn project supported by Inland Norway University of Applied Sciences and a grant from The Research Council of Norway (NFR project 221056; https://prosjektbanken.forskningsradet.no/#/project/NFR/221056/Sprak=en) to HPA. Work done by MSG was only partly funded. The Research Council of Norway (https://www.forskningsradet.no/en/) had no role in study design, data collection and analysis, decision to publish, or preparation of the manuscript.

**Competing interests:** The authors have declared that no competing interests exist.

The production of chemical defense metabolites is one of many defense strategies used by plants to minimize the negative effect of herbivory on plant fitness [20, 21].

Several hypotheses about constitutive and inducible defense are relevant for bilberry-herbivore interactions. In this paper we use 'constitutive defense' and 'inducible defense' as Tuomi and colleagues do [22]: constitutive defense levels are not affected by herbivores, whereas induced defense refers to the change in plant resistance as a response to herbivory. Induced defense is only possible if the plant possesses phenotypic plasticity in defense, which applies to bilberry [23]. Below, we introduce briefly three existing plant defense hypotheses and describe how plant nutrient concentration and defenses are expected to be influenced by soil productivity, light availability and herbivory. After combining this information (Fig 1) we present our own predictions.

The Optimal Defense (OD) hypotheses state that defenses are costly (in terms of fitness) because they divert resources from growth, and assume that herbivory is the primary selective force shaping quantitative patterns of secondary metabolism. As a result, expression of resistance (e.g., production of inducible defenses, which are secondary metabolism compounds) should be low when herbivores are nearly absent and increase when the plant is under attack [24–29].

The Carbon:Nutrient Balance (CNB) hypothesis is a model of how the supply of carbon and nutrients in the environment influences the phenotypic expression of secondary metabolism by plants [22, 28, 30]. The CNB hypothesis predicts that increased nitrogen availability permits plants to allocate more carbon to growth, resulting in less carbon-based defense compounds (CBDCs). A similar decrease in CBDCs is predicted with increasing shade, as this decreases the C:N ratio by limiting carbon assimilation more than nutrient uptake [31]. Accordingly, light availability is positively correlated with production of many CBDCs [19, 28, 32, 33]. Furthermore, herbivory can alter the carbon:nutrient balance within plants, that may influence the level of CBDCs. Because many deciduous woody species growing on nutrient-poor soils store carbon in stems and roots [22, 34], herbivory on shoots and leaves is expected to increase the level of CBDCs in bilberry.

The expanded Growth-Differentiation Balance (GDB) hypothesis includes all extrinsic factors affecting secondary metabolism, not only carbon and nutrients as in the CNB hypothesis. The GDB hypothesis acknowledges that in plant development there is a constant tradeoff between growth and differentiation requirements. For any resource-shortage that slows growth more than it slows photosynthesis, the GDB hypothesis predicts a unimodal effect of availability of this resource on secondary metabolite production [28, 31, 35]. Consequently, under non-shady and low soil productivity conditions, nitrogen-demanding growth processes are more limited than production of CBDCs.

In addition to concentration and type of defense compounds, palatability is also affected by nutrient concentration in plants [36, 37]. Nitrogen concentration, which is often used as a proxy for nutrient concentration, increases in bilberry after nitrogen fertilization and is positively related to habitat productivity [19, 38–43, but see 44, 45]. Nitrogen concentration in leaves is negatively related to light availability [46–48, see also 49]. Pruning (partial or complete removal of stem/shoots) reduces bud numbers and increases the root:shoot ratio, resulting in decreased competition for nutrients among meristems and, thus, increased nutrient concentration in new plant tissue [50–54]. Indeed, nitrogen concentration increases after browsing in several woody species, often regardless of soil productivity [55–59].

Based on what precedes, palatability of bilberry leaves to insect herbivores is affected by the combined concentration in CBDCs and nutrients, which are affected by light availability, soil productivity and herbivory (Fig 1). According to Fig 1I, the relationship between mammalian herbivory and subsequent insect herbivory on bilberry varies depending on whether insects

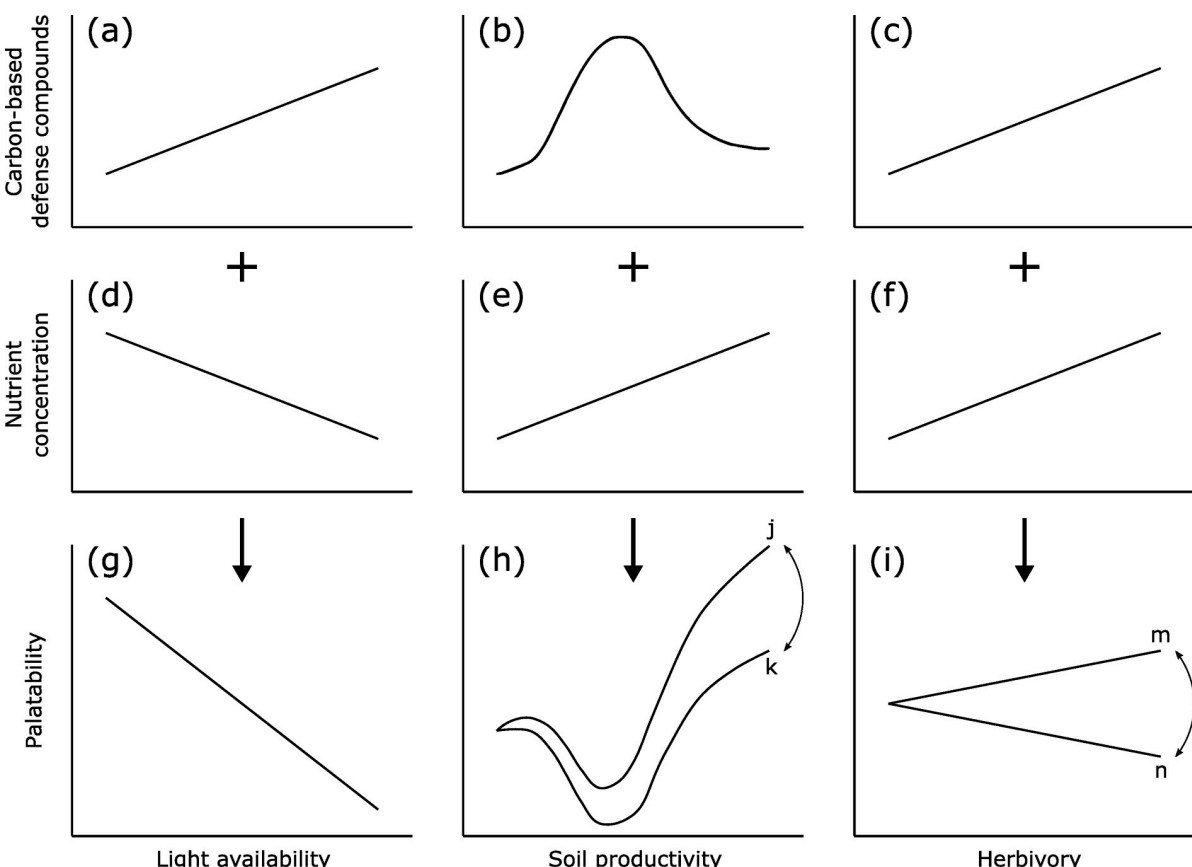

**Fig 1. Light, soil and herbivory affecting bilberry defense compounds and nutrients = bilberry palatability.** Theoretical relationship between (a,d,g) light availability, (b,e,h) soil productivity and (c,f,i) herbivory (predictor variables) and (a,b,c) carbon-based defense compounds (CBDCs), (d,e,f) nutrient concentration and (g,h,i) palatability (response variables) in/of bilberry (*Vaccinium myrtillus*) leaves, assuming a positive linear relationship between secondary metabolites and CBDCs. The combination (indicated by a plus sign) of (a) and (d) results (indicated by a vertical arrow) in (g); (b) combined with (e) results in (h); (c) combined with (f) results in (i). Herbivory refers to previous mammalian herbivory (pruning). Palatability is the combined effect of CBDCs and nutrient concentration. Palatability under different (h) soil productivity (level of nutrients available to the individual plant) and (i) herbivory pressure ranges between (h) j and k and (i) m and n, dependent on whether palatability is less (j and m) or more (k and n) affected by CBDCs than by nutrient concentration. Sources (a-c): (a) Carbon:Nutrient Balance (CNB) hypothesis, Growth-Differentiation Balance (GDB) hypothesis; (b) GDB hypothesis; (c) Optimal Defense hypotheses, CNB hypothesis. Sources (d-f): see references in text.

profit more from increasing nutrient concentrations in bilberry leaves than they suffer from increasing defense compound concentrations in these leaves, or vice versa. While this relationship has been studied in other woody species, e.g., northern willow (*Salix glauca* L.) [58], this relationship is, to our knowledge, not known for bilberry. The aim of our study was to assess whether bilberry leaf palatability to insects is affected by light availability, soil productivity and previous mammalian herbivory. Therefore, we investigated bilberry leaf palatability to insects under different levels of light availability, soil productivity and mammalian herbivory pressure in small sampling units (1m x 1m) in six boreal forest areas in southeastern Norway in the period 2013–2015. We assumed that under similar light availability and soil productivity conditions, the change in leaf palatability caused by induced changes in CBDCs in leaves is counter-balanced by induced changes in nutrient concentration in these leaves, resulting in bilberry leaf palatability showing no correlation with previous mammalian herbivory. Based on this assumption, and the theory highlighted in Fig 1G–1I we predicted that bilberry leaf palatability:

I. is negatively correlated with light availability,

II. shows a unimodal relationship with soil productivity, and

III. is not correlated with previous mammalian herbivory.

## Methods

### Study area

We conducted the study in the Østerdalen valley in southeastern Norway (Fig 2) in the period 2013–2015. The study area was at elevation 288–810 m a.s.l. and consisted mainly of coniferous boreal forest interspersed with streams, marshes and grasslands with free-ranging domestic livestock (sheep and cows) during the snow-free season. Common wild mammalian herbivore species in the area were moose (*Alces alces* L.), red deer (*Cervus elaphus* L.), roe deer (*Capreolus capreolus* L.), mountain hare (*Lepus timidus* L.), several small rodent species and, to the west of the Glomma River (Fig 2), reindeer (*Rangifer tarandus* L.). During the study period, annual mean temperature was 4˚C (-9˚C in January and 16˚C in July) at Evenstad weather station (61˚26'N, 11˚05'E, elevation 257 m a.s.l.); annual mean precipitation was 818 mm at Rena Flyplass weather station (61˚11'N, 11˚22'E, elevation 255 m a.s.l.) [60]. No permits for field site access were necessary, according to Norwegian law (friluftsloven: LOV-1957-06-28-16) that permitted access by foot to natural areas.

### Study design

We sampled within six blocks of 16 km$^2$ (4 km x 4 km) each (Fig 2, black squares). In the center of each block, we used four 1.5 km long transect lines, parallel and spaced by 500 m. Each line contained four survey locations for bilberry data collection and soil sampling. Each survey location consisted of two vegetation sampling quadrats of 1 m$^2$ (1m x 1m, permanently marked), separated by approximately 40 m. In each quadrat we estimated bilberry cover (%) and insect herbivory on bilberry leaves: we estimated chewing damage as the proportion of leaf area eaten in the shape of holes ('hole herbivory') and the proportion of leaf area eaten at the edge of the leaves ('edge herbivory'). We also looked for signs of 'present-year mammalian herbivory', i.e., herbivory on stems and shoots that had occurred during or since the previous winter, and we estimated the proportion of biomass that had been taken away (refered to as 'previous mammalian herbivory' in this paper). We sampled the vegetation once a year (8–29 July 2013, 1–22 July 2014 and 7–24 July 2015). In 2015 we recorded tree species composition of the surrounding forest for each quadrat. We estimated the proportion of shade from the tree canopies at each survey location in 2014 and used these for both quadrats. We used three categories: less than 20% (low shade level), between 20% and 80% (intermediate shade level), more than 80% (high shade level). As low and high shade levels were assumed to result in clearly different palatability (Fig 1G), these categories were made narrow. We collected soil samples (the upper organic layer down to maximum 10 cm) nearby every quadrat with a metal bulb planter in October 2014. We merged the two soil samples at each survey location and used these for both quadrats. We stored the samples frozen (-18˚C) prior to analysis. All samples were analyzed for ammonium lactate extractable phosphorus (measuring method uncertainty ± 20%, method reference SS028310T1/SS-EN) and total nitrogen (measuring method uncertainty ± 10%, method reference EN 15104:2011/EN 15407:2011) (Eurofins Food & Agro Testing Sweden AB, Kristianstad/Linköping, April 2015). We did not use inorganic ammonium ($NH_4^+$) or nitrate ($NO_3^-$) concentrations, as organic nitrogen is an important source of nitrogen to bilberry [45, see also 61].

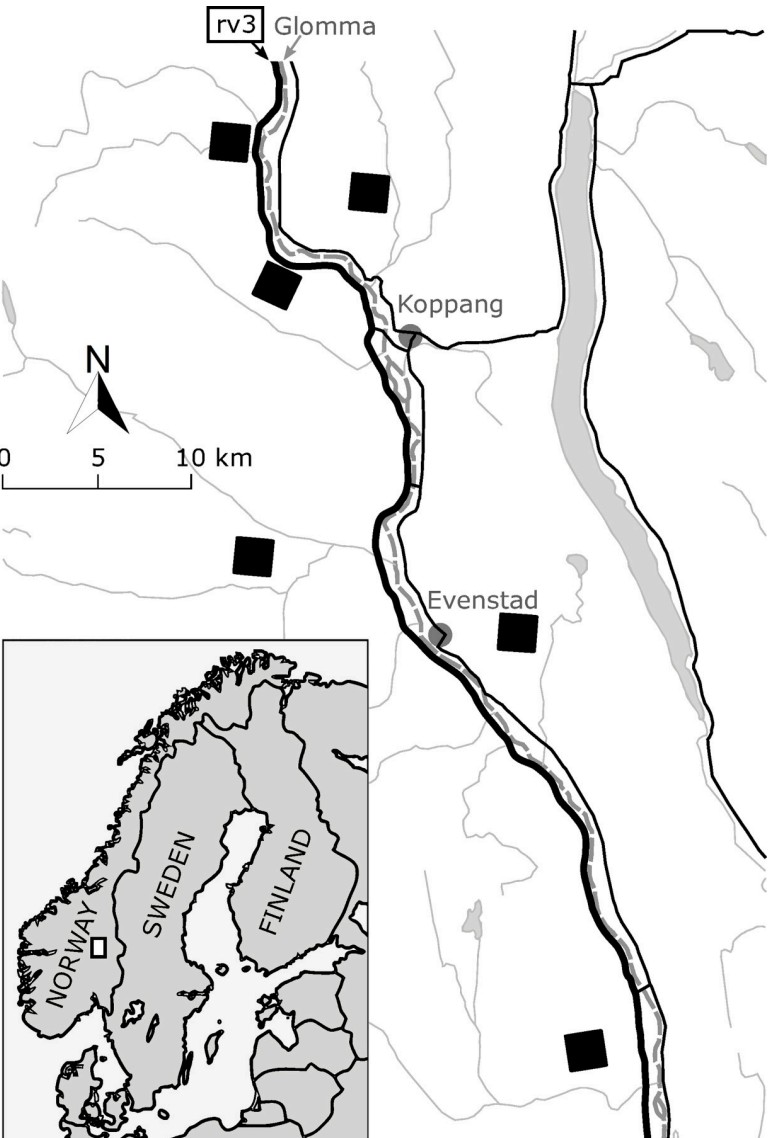

**Fig 2. Map of the study area (inset: White square) in southeastern Norway.** Thick black line: main road 3 (rv3); dashed grey line: the Glomma River; black squares: sampling blocks, see text.

## Data analyses

We focused our analyses on quadrats in evergreen forest where bilberry was present, and with non-missing data for insect and mammalian herbivory, shade, phosphorus and nitrogen, which left 455 quadrats for analyses. We considered that the single sampling for shade and soil was representative for the whole study period. We combined edge and hole herbivory as 'insect herbivory' (response variable), which we used as a proxy for bilberry leaf palatability.

**Prediction I: Leaf palatability and light availability.** To investigate prediction I we modeled insect herbivory as a function of shade ('shade model'). We added year as a fixed effect to account for annual variability, caused by variability in, e.g., field workers, vole density and weather (S1 File). The response variable (insect herbivory) was a proportion (a continuous variable with a value from 0 to 1), therefore we used a beta distribution with a logit link function

[62]. We used generalized linear mixed models (GLMMs) fitted using the package 'glmmTMB' in the software 'R' [63]. All analyses in this study were performed in R, version 3.6.2. [64]. Before modeling, we checked for collinearity between all predictor variables. Prior to analyses we used the transformation: (insect herbivory x (n– 1) + 0.5) / n (where n is the number of observations) for the response variable, to deal with actual observations equal to 0 or 1 [65, 66]. Given our study design, we initially included a nested random component in the model with survey location nested within transect line nested within block. The corresponding estimates of variance were very small so we removed the line and block grouping variables from the random component in the models that we used for analyses [67]. We used Akaike's information criterion (AIC) to compare the shade model with a similar model without the fixed effect shade [68]. We used the package 'emmeans' to further investigate the relationship between insect herbivory and light availability [69]. We validated the model by evaluating residual diagnostics using the package 'DHARMa' [70]. Unless otherwise stated, we used a significance level of 5% in all our analyses in this study.

**Prediction II: Leaf palatability and soil productivity.** To investigate prediction II we first performed a principal component analysis (PCA) with the two standardized variables phosphorus and nitrogen to obtain a single composite covariate (called PC1) for soil productivity to use in our subsequent modeling (S1 File) [71]. We modeled insect herbivory as a function of soil productivity ('soil model'). Similar to the shade model (see Prediction I), we first checked for collinearity between all predictor variables, used the transformed response variable and a beta distribution with a logit link function, added year as a fixed effect and used survey location in the random component. We used generalized additive mixed models (GAMMs) fitted using the package 'mgcv' [72, 73]. To evaluate the existence of a unimodal relationship we compared a soil model fitted with a non-linear relationship, i.e., a GAMM with a smooth term, with a soil model fitted with a linear relationship, i.e., a GAMM without a smooth term, and with a similar model without the fixed effect soil productivity, by their AIC values. We validated the models by evaluating the standardized residuals graphically [74–76].

**Prediction III: Leaf palatability and mammalian herbivory.** To investigate the relationship insect herbivory–previous mammalian herbivory under different levels of light availability and soil productivity, we made scatterplots and added linear regression lines for insect and previous mammalian herbivory, at all possible combinations of shade conditions and soil productivity levels. Soil productivity levels were obtained by categorizing the numerical variable soil productivity (see prediction II) into three evenly distributed (same number of observations) soil productivity classes: low, intermediate, high. Means and standard error (SE) values for the different classes were calculated with the package 'emmeans'. To investigate prediction III, we modeled insect herbivory as a function of previous mammalian herbivory, shade, soil productivity, and all their possible interactions. Similar to the shade model (see Prediction I), we first checked for collinearity between all predictor variables, used GLMMs fitted using the package 'glmmTMB', used the transformed response variable and a beta distribution with a logit link function, added year as a fixed effect and used survey location in the random component. In addition, we standardized the variable previous mammalian herbivory prior to modeling using the package 'arm' [77]. We performed model selection using AIC and model evaluation using the package 'DHARMa'. We used the parameter estimates of the best (most parsimonious) model to predict insect herbivory on bilberry leaves following mammalian herbivory under different light conditions (low, intermediate, high shade levels) and on soils with different productivity (low, intermediate, high levels). To visualize the predicted insect herbivory values we used the first, second (the median) and third quartile of the variable soil productivity for low, intermediate and high soil productivity, respectively.

## Results

Across quadrats, herbivory was low (Fig 3) but frequent, and more often due to insects than to mammals (S1 File). Quadrats at exposed and half-open locations were twice as frequent as quadrats in shady conditions, nitrogen and phosphorus levels showed little variation (S1 File).

### Prediction I: Leaf palatability and light availability

Correlation between the predictor variables shade and year was very low (Pearson's correlation coefficient: $\rho = 0.027$). Estimates of variance for the initially used nested random component were: survey location:(line:block) = 0.008 (n = 88), line:block = 0.004 (n = 24), block = 0.007 (n = 6). The estimated variance of survey location, when only survey location was used in the random component, was 0.021 (n = 88). The shade model had a lower AIC value than a similar model without the fixed effect shade (but with the fixed effect year and the random component): -2605.23 versus -2598.80, respectively. The shade model revealed a positive correlation between insect herbivory and shade (S1 and S3 Tables). Using the shade model, there was no significant difference in insect herbivory between the intermediate and high shade levels (Tukey's HSD test: df = 448, $P = 0.68$). In this model, insect herbivory at low shade level differed from insect herbivory at intermediate and high shade levels (Tukey's HSD test: df = 448, $P = 0.02$ and 0.01, respectively). Analyses with the DHARMa package showed that the model fit was quite poor (e.g., overdispersion present). In a linear model (both with and without the fixed effect year; no random component, Gaussian distribution with identity link) insect herbivory at high shade levels differed from insect herbivory at both low and intermediate shade levels but there was no significant difference between low and intermediate shade levels (Fig 3A; S2 Table). We conclude that our data supports prediction I.

### Prediction II: Leaf palatability and soil productivity

Correlation between the predictor variables soil and year was low (Pearson's correlation coefficient: $\rho = 0.092$). Estimates of variance for the initially used nested random component in the soil model fitted with a non-linear relationship were: survey location:(line:block) = 0.144 (n = 88), line:block = 0.024 (n = 24), block = 0.028 (n = 6). The estimated variance of survey location, when only survey location was used in the random component, was 0.194 (n = 88). Based on their AIC values, there was no difference between the soil model with a non-linear fit, the soil model with a linear fit and a similar model without the fixed effect soil productivity (but with the fixed effect year and the random component) (AIC values: 1492.151, 1490.151 and 1491.687, respectively). Consequently, there was not enough non-linearity in the relationship between the variables to warrant a (more complex) non-linear model. Moreover, based on the AIC values the model with and the model without the covariate soil were as good in predicting insect herbivory. Model validation showed that the model fit for all models was quite poor (e.g., heterogeneity present). We conclude that our data does not support prediction II.

### Prediction III: Leaf palatability and mammalian herbivory

Observations with the 151 highest PC1 values were assigned to class 'low soil productivity levels' (mean = 1.193, SE = 0.035), observations with the 151 lowest PC1 values were assigned to class 'high soil productivity levels' (mean = -1.156, SE = 0.035), the remaining ones (n = 153) were assigned to class 'intermediate soil productivity levels' (mean = -0.037, SE = 0.035). Insect herbivory related to previous mammalian herbivory is shown in Fig 4, at all possible combinations of shade conditions and soil productivity levels. In six panels no correlation (P > 0.10) was present between mammalian and subsequent insect herbivory. The combination

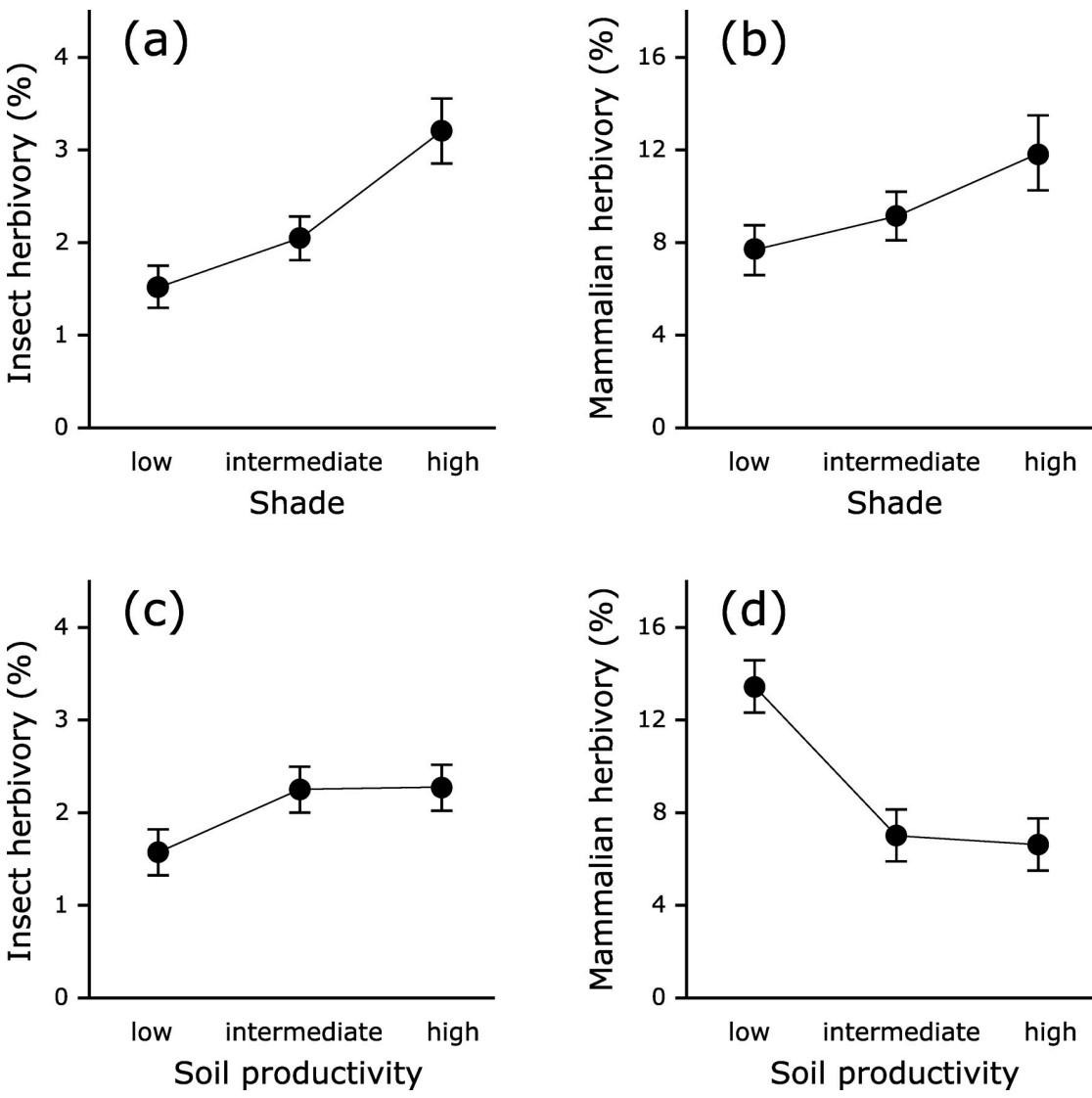

**Fig 3. Insect and mammalian herbivory on bilberry at different light availability and soil productivity levels.** Estimated herbivory by insects (a,c) and mammals (previous mammalian herbivory) (b,d) on bilberry (*Vaccinium myrtillus*) per light availability class (a) and soil productivity class (b) over the whole study period. Means ± SE. n = 455.

intermediate shade levels and low productivity levels yielded a tendency for a negative linear relationship (slope = -0.02, ANOVA: $F_{1,53}$ = 2.85, $P$ = 0.097) but the explained variation was low (adjusted R-squared = 0.03). A significant positive linear relationship between mammalian and subsequent insect herbivory was present in two panels: high shade levels and intermediate productivity levels (slope = 0.19, ANOVA: $F_{1,24}$ = 10.85, $P$ = 0.0031) and intermediate shade levels and high productivity levels (slope = 0.13, ANOVA: $F_{1,57}$ = 9.97, $P$ = 0.0025). The explained variation was 28% and 13%, which can be considered as important effect sizes in ecological studies with a low degree of control, like our field study [78, 79]. The linear relationship between insect herbivory and previous mammalian herbivory independent of shade and soil was significant (slope = 0.04, ANOVA: $F_{1,453}$ = 12.46, $P$ = 0.00046) but the explained variation was very low (adjusted R-squared = 0.02), see S1 File.

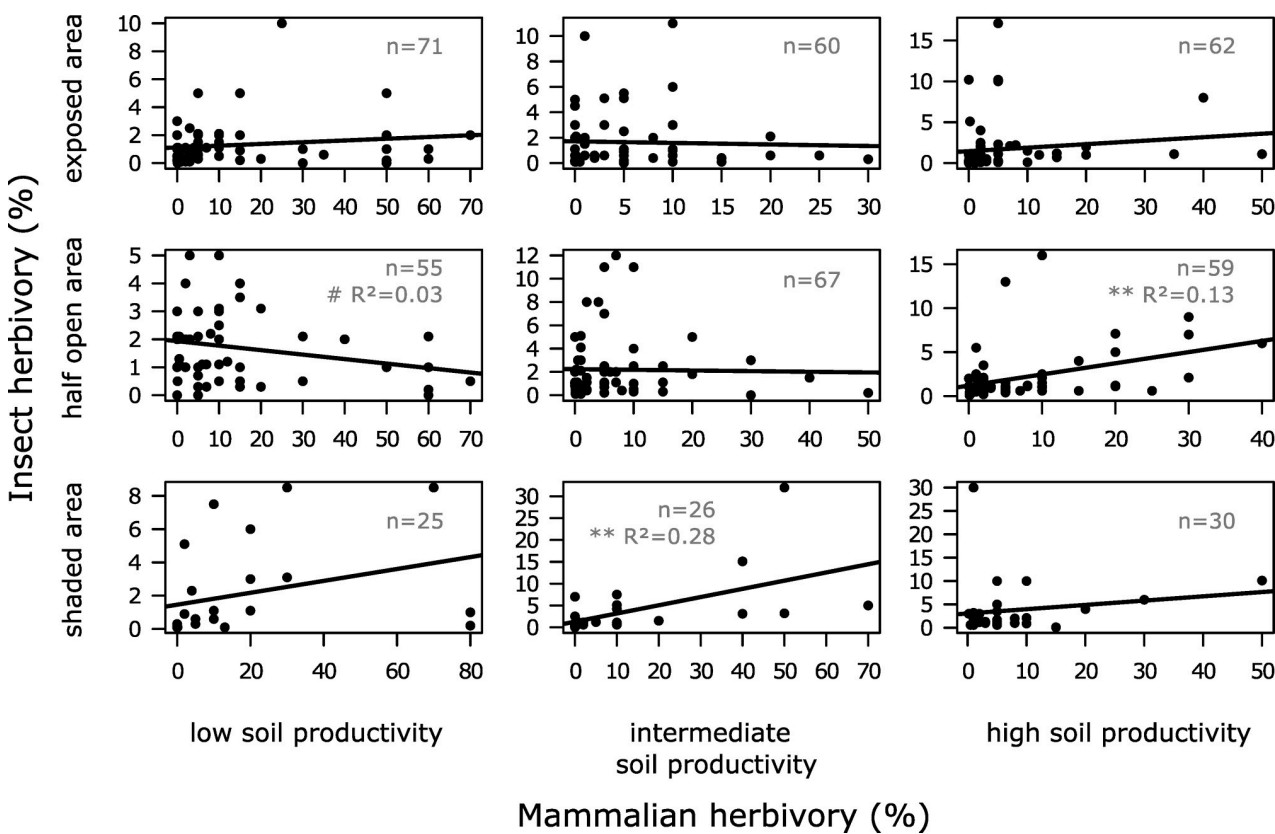

**Fig 4. Observed insect versus mammalian herbivory on bilberry at different light availability and soil productivity levels.** Insect herbivory versus previous mammalian herbivory on bilberry (*Vaccinium myrtillus*) in exposed areas (shade < 20%), half open areas (shade between 20% and 80%) and shaded areas (> 80% shade) and at low, intermediate and high soil productivity levels. In each panel the number of observations (n) is given, and the *P*-value (ANOVA) for the linear regression (regression line shown in each panel) is indicated as: *** $0 < P < 0.001$; ** $0.001 < P < 0.01$; * $0.01 < P < 0.05$; # $0.05 < P < 0.10$; blank $P > 0.10$. If $P < 0.10$ the adjusted R-squared value is given.

Correlation in the set of predictor variables that we used in our modeling was low (highest Pearson's correlation coefficient: $\rho = 0.26$). Estimates of variance for the initially used nested random component in the full model were: survey location:(line:block) = 0.018 (n = 88), line: block = 0.003 (n = 24), block = 0.010 (n = 6). The estimated variance of survey location in the full model, when only survey location was used in the random component, was 0.032 (n = 88). In total 35 models were used in our model selection analyses. The best model included interactions between soil productivity and previous mammalian herbivory, between shade and previous mammalian herbivory, and between shade and soil productivity (S4 Table). Analyses with the DHARMa package showed that the model fit was quite poor (e.g., overdispersion present). The parameter estimates of the best model (Table 1) were used to predict insect herbivory in Fig 5. These predicted values are for 2014 as this year had the highest estimate compared to 2013 and 2015 (Table 1); predictions for 2013 and 2015 show similar curves but with less amplitude. In shady conditions, predicted insect herbivory increased with previous mammalian herbivory; the rate of increase was lowest at low soil productivity levels and highest at high soil productivity levels (Fig 5C). With low and intermediate shade levels (Fig 5A and 5B), insect herbivory was predicted to show either little positive correlation with previous mammalian herbivory (soils with high productivity levels) or (almost) no correlation (soils with low and intermediate productivity levels). We conclude that our data does not support prediction III.

**Table 1. Parameter estimates for the best model of variables affecting insect herbivory on bilberry leaves.** Parameter estimate, standard error, 95% confidence interval and *P*-value are presented for the intercept and each of the fixed effects in the best model (S4 Table). Note that the model used a logit link function (the estimates are on a logit-scale, not the response scale), that the predictor variable previous mammalian herbivory was standardized and that the response variable was transformed prior to analyses (see text). Therefore, also the back-transformed estimate (back-transformed from both logit transformation and response variable transformation) is presented for the intercept and each of the fixed effects (thus, this value is on the response scale). Parameter estimate and 95% confidence interval are also presented for the standard deviation of the random component and for the dispersion parameter. Number of observations: 455.

| Parameter | Estimate | SE | lCI | uCI | *P*-value | Sign | BE |
|---|---|---|---|---|---|---|---|
| Intercept | -3.92 | 0.09 | -4.11 | -3.74 | 0.00 | *** | 0.02 |
| Soil | -0.06 | 0.07 | -0.19 | 0.08 | 0.40 | | 0.49 |
| Shade < 20% | -0.23 | 0.09 | -0.41 | -0.05 | 0.01 | * | 0.44 |
| Shade > 80% | 0.07 | 0.11 | -0.15 | 0.30 | 0.52 | | 0.52 |
| Mammal | 0.12 | 0.13 | -0.14 | 0.38 | 0.35 | | 0.53 |
| Year 2014 | 0.42 | 0.09 | 0.24 | 0.60 | 0.00 | *** | 0.60 |
| Year 2015 | -0.05 | 0.10 | -0.23 | 0.14 | 0.63 | | 0.49 |
| Soil : Mammal | -0.24 | 0.06 | -0.36 | -0.13 | 0.00 | *** | 0.44 |
| Soil : Shade < 20% | 0.05 | 0.09 | -0.12 | 0.23 | 0.54 | | 0.51 |
| Soil : Shade > 80% | -0.24 | 0.12 | -0.46 | -0.01 | 0.04 | * | 0.44 |
| Shade < 20% : Mammal | 0.20 | 0.18 | -0.16 | 0.56 | 0.27 | | 0.55 |
| Shade > 80% : Mammal | 0.69 | 0.16 | 0.37 | 1.01 | 0.00 | *** | 0.67 |
| Location (st.dev.) | 0.18 | | 0.08 | 0.37 | | | |
| Dispersion parameter | 55.83 | | 47.33 | 65.85 | | | |

Mammal = previous mammalian herbivory; Location (st.dev) = survey location (standard deviation); SE = standard error; lCI = lower 95% confidence interval;

uCI = upper 95% confidence interval; Sign = significance level

*** 0 < *P* < 0.001

** 0.001 < *P* < 0.01

* 0.01 < *P* < 0.05; blank *P* > 0.05; BE = back-transformed estimate.

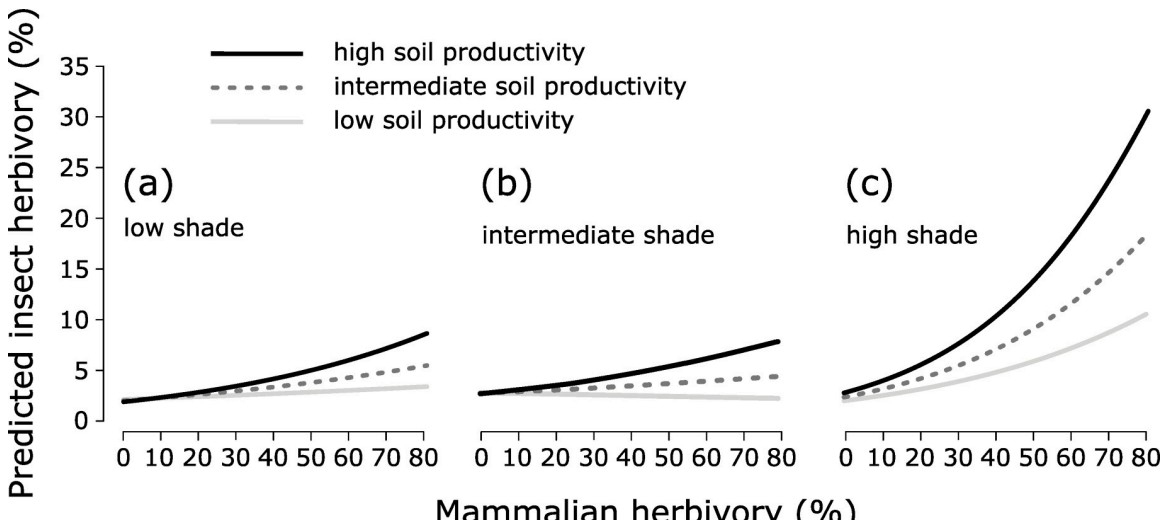

**Fig 5. Predicted insect herbivory on bilberry leaves following mammalian herbivory.** Predicted insect herbivory on bilberry (*Vaccinium myrtillus*) leaves, growing at low, intermediate and high soil productivity levels, as a function of previous mammalian herbivory in 2014, under conditions of (a) low, (b) intermediate and (c) high shade levels. Predictions based on parameter estimates for the best model (Table 1).

## Discussion

Plant defense theory predicts that palatability of bilberry leaves to insect herbivores is influenced by light availability, soil productivity and herbivory by mammals, as these factors affect nutrient and CBDC concentration (Fig 1). We found that insect herbivory had a positive relationship with previous mammalian herbivory at high shade levels. At intermediate and low shade levels, this relationship was weak (bilberry growing at high soil productivity levels) or absent (bilberry growing at intermediate or low soil productivity levels) (Fig 5).

### Prediction I: Leaf palatability and light availability

According to our model, insect herbivory increases with previous mammalian herbivory and with soil productivity, especially under shady conditions (Fig 5). The difference between the observed relationship insect herbivory–soil productivity (Fig 3C) and the model predictions (Fig 5) indicates that also previous mammalian herbivory and light conditions are influencing palatability of bilberry leaves to insects, in accordance with the mentioned theories on plant defense. Richardson and colleagues [44] found that insect herbivory on bilberry increased after nutrient addition and with experimental warming. However, they used open top chambers (OTCs) in which photosynthesis often is reduced [80]. This means that the increase in insect herbivory found by Richardson and colleagues [44] may have been caused by a combination of fertilization, higher temperature and reduced light availability. The latter is in accordance with our finding that light availability is important for leaf palatability and in line with our prediction I.

Our results indicate that light availability is more important for variation in bilberry leaf palatability than soil nutrient conditions. This is in agreement with the results from a study on bilberry leaves in northern Finland [81]. In a study in northern Sweden, some particular CBDCs (flavonoids) in bilberry leaves were not affected by nitrogen fertilization [82]. The authors suggest that light conditions may be a regulator for the synthesis and accumulation of flavonoids, which are important in plant protection against ultraviolet-B radiation (UV-B) [82, 83]. In our study we incorporated light conditions by estimating the proportion of shade but we did not measure temperature nor UV-B, which affect several CBDCs in foliage, potentially altering insect herbivore performance either positively or negatively [84–92].

### Prediction II: Leaf palatability and soil productivity

We found no support for a non-linear effect of soil productivity on bilberry leaf palatability to insects. Our observations may not cover the full ecological range for soil productivity of bilberry (S1 File). Soil productivity is generally low in boreal forests [93, 94]. The productivity might be too low to see any response in insect herbivory on bilberry. Additionally, the small spatial scale of the study (one valley) may have limited the spatial variation in soil productivity. Indeed, there is only small variation in nitrogen and phosphorus concentrations in our dataset. Furthermore, as bilberry is adapted to relatively nutrient-poor environments, increased soil productivity may not trigger a direct response [95].

The CNB hypothesis predicts that nitrogen enrichment permits plants to allocate more carbon to growth, resulting in a decrease in CBDCs. This does not apply to all plant secondary metabolites, as proteins and many phenolics compete for the precursor phenylalanine [96, 97]. This precursor was used by Jones and Hartley [98] in their protein competition model for predicting total phenolic concentration in leaves. Consequently, as biosynthesis of terpenoids and of hydrolyzable tannins presumably proceeds without direct competition with protein synthesis [96], these secondary metabolites are likely to follow the CNB hypothesis, while others, e.g., flavonoids [19] and condensed tannins, may not. Therefore, if bilberry is attacked by leaf-

chewing insect species that are less sensitive to terpenoids and hydrolyzable tannins but that respond negatively to flavonoids and condensed tannins in leaves, insect herbivory on bilberry leaves may not be correlated to soil productivity. This means that the relationship between leaf palatability and soil productivity may depend on the insect species involved.

### Prediction III: Leaf palatability and mammalian herbivory

An important limitation of our study is the uncertainty in our main covariate: the estimation of the proportion of biomass that had been taken away by mammals. Estimating something that is no longer present can be challenging! We did not take any observations of biomass before herbivory, for example by using photographs [99]. Still, we found limited support for a significant positive relationship between observed mammalian and subsequent insect herbivory. At high shade levels the predicted insect herbivory increased with increasing previous mammalian herbivory. At low and intermediate shade levels, our third prediction seems to hold at least for bilberry growing at low and intermediate soil productivity levels.

The observed increase in insect herbivory following mammalian herbivory indicates that under certain light and soil nutrient conditions bilberry leaf palatability is more affected by leaf nutrient concentration than by leaf CBDC concentration (m in Fig 1I). As reviewed by Koricheva and colleagues, several studies, including a study with leaf-eating larvae on bilberry, showed that insect performance on experimentally stressed woody species improved with stress level until reaching some threshold, above which performance declined [23, 100]. However, a non-linear model with soil production did not markedly improve the predictions and was selected against with our current data.

### Conclusion

Our study indicates that light availability is important for bilberry leaf palatability, as insect herbivory on bilberry leaves increased with increasing shade (confirming our first prediction). Our results indicate that under certain light and soil nutrient conditions bilberry leaf palatability following mammalian herbivory on bilberry is more affected by leaf nutrient concentration than by leaf CBDC concentration. Furthermore, we did not find a straightforward correlation between insect herbivory and soil productivity alone (falsifying our second prediction), without taking into account light conditions: our results indicate that at high shade levels bilberry leaf palatability is positively correlated with previous mammalian herbivory (falsifying our third prediction) and this response is magnified at higher soil productivity levels. At low to intermediate shade levels, this response is only present under high soil productivity levels. Our results indicate that light availability is more important for variation in bilberry leaf palatability than soil productivity.

### Supporting information

**S1 File.**
(PDF)

**S1 Table. Insect herbivory: Estimated Marginal Means (EMMs) per shade level in the shade model.** Estimated marginal mean values for insect herbivory, their standard error, degrees of freedom and 95% confidence intervals are presented for each level of the variable shade in the shade model. Note that the model used a logit link function (the estimates are on a logit-scale, not the response scale) and that the response variable was transformed prior to analyses (see text in the manuscript). Therefore, also the back-transformed estimated marginal means (back-transformed from both logit transformation and response variable

transformation) are presented (thus, these values are on the response scale). To make differences visible, three digits are given for the back-transformed estimated marginal means. Results are averaged over the levels of the variable year. Number of observations: 455.
(PDF)

**S2 Table. Insect herbivory: Estimated Marginal Means (EMMs) per shade level in a linear model and the contrast estimates with Tukey's HSD test values.** Estimated marginal mean values for insect herbivory, their standard error, degrees of freedom and 95% confidence intervals are presented for each level of the variable shade in a linear model with and without year as a fixed effect. In the first model, results are averaged over the levels of the variable year. Similar information is presented for the contrast estimates; in addition, their *P* values based on Tukey's HSD test are given. Number of observations: 455.
(PDF)

**S3 Table. Parameter estimates for the shade model of variables affecting insect herbivory on bilberry leaves.** Parameter estimates, standard error, 95% confidence interval and *P*-values are presented for the intercept and each of the fixed effects in the shade model. Note that the model used a logit link function (the estimates are on a logit-scale, not the response scale) and that the response variable was transformed prior to analyses (see text in the manuscript). Therefore, also the back-transformed estimates (back-transformed from both logit transformation and response variable transformation) are presented for the intercept and each of the fixed effects (thus, these values are on the response scale). Parameter estimates and 95% confidence interval are also presented for the standard deviation of the random component and for the dispersion parameter. Number of observations: 455.
(PDF)

**S4 Table. Modeling of variables affecting insect herbivory on bilberry leaves in southeastern Norway in 2013–2015.** Model inferences based on Generalised Linear Mixed Modeling (beta regression with logit link). Best models based on AIC selection; only the five best models are presented, and the null model for comparison purposes. In addition to the presented model sets in the table, all models contained the random component (1|survey location). The first model (Δ AIC = 0.00) is the most parimonious model. The second model (Δ AIC = 0.50) is the full model. See text for description of the fixed effects and random component. Number of observations: 455.
(PDF)

## Acknowledgments

We thank Olivier Devineau for invaluable help with statistics and extensive comments on the manuscript, David Carricondo Sánchez for coordinating and conducting fieldwork, help with data analyses and for constructive comments, Barbora Malá for substantially contributing to the study design, conducting additional fieldwork and analysing data, Cyril Milleret for help with R and ArcGIS, Morten Odden and Lasse Asmyhr for help with organising and coordinating fieldwork, Maria Greger, Jo Inge Breisjøberget, Barbara Zimmermann, Ane Eriksen Hamilton, Gitte C. Kloek, Vladimir Naumov and Antonio B. S. Poléo for scientific and editorial advice, Zea Walton and Annie Loosen for help with some language issues, Marieke Gonlag-Schrijvers for editorial advice and Yagya Raj Bhatt and Bernardo Toledo González for punching data. We thank all fieldworkers for their efforts: Clementine Crombez, Jason Kandume, Jorge Galindo Guiterrez, Julie Saez, Siegfrid Waas, Solenne Lheritier, Jakob K. N. Brunner, Melissa Brilman, Matthieu Gibert, Carole Parrel, Vincent Hetter, Scarlet van Os, Sabrina

Dietz, Axel Becker, Timo Förster, Barbara Joncour, Julia Gómez Catasús, Magnus Hoff Olsen, Farina Sooth, Thomas Vogler, Sofia Willebrand, Emelie Önstedt, Umer Qureshi, Claire Tachon, Olivier Duchene, Andreja Kovše, Jaka Tegelj, Sašo Veselinovič, Andreas Hein, Audrey Jansseune, Marieke Gehem, Florian Nöscher, Vincent Baudon, Corentin Bouffanet, Jan Kiehne, Pierre Lequay, Petrus Martiskin, Urška Mrak, Falk Schreiner, Ulvi Selgis, Md. Shamsuzzaman and all others not mentioned here. MSG thanks the Stack Exchange Q&A web communities Stack Overflow and Cross Validated for invaluable statistical and analytical insights. MSG and CS especially thank Harry P. Andreassen, who, no longer with us († 21 May 2019), initiated this study and contributed highly to earlier versions of the manuscript.

## Author Contributions

**Conceptualization:** Christina Skarpe, Harry Peter Andreassen.

**Data curation:** Marcel Schrijvers-Gonlag.

**Formal analysis:** Marcel Schrijvers-Gonlag, Harry Peter Andreassen.

**Funding acquisition:** Harry Peter Andreassen.

**Investigation:** Marcel Schrijvers-Gonlag, Christina Skarpe.

**Methodology:** Marcel Schrijvers-Gonlag, Christina Skarpe, Harry Peter Andreassen.

**Project administration:** Marcel Schrijvers-Gonlag.

**Supervision:** Christina Skarpe, Harry Peter Andreassen.

**Visualization:** Marcel Schrijvers-Gonlag.

**Writing – original draft:** Marcel Schrijvers-Gonlag.

**Writing – review & editing:** Marcel Schrijvers-Gonlag, Christina Skarpe, Harry Peter Andreassen.

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
