## [Decision Letter · Decision Letter 0]

24 Oct 2019

PONE-D-19-24380

Influence of light availability and soil productivity on insect herbivory on bilberry (Vaccinium myrtillus L.) leaves following mammalian herbivory

PLOS ONE

Dear Mr. Schrijvers-Gonlag,

Thank you for submitting your manuscript to PLOS ONE. After careful consideration, we feel that it has merit but does not fully meet PLOS ONE’s publication criteria as it currently stands. Therefore, we invite you to submit a revised version of the manuscript that addresses the points raised during the review process.

We would appreciate receiving your revised manuscript by Dec 08 2019 11:59PM. To enhance the reproducibility of your results, we recommend that if applicable you deposit your laboratory protocols in protocols.io, where a protocol can be assigned its own identifier (DOI) such that it can be cited independently in the future. For instructions see: http://journals.plos.org/plosone/s/submission-guidelines#loc-laboratory-protocols

We look forward to receiving your revised manuscript.

Kind regards,

Livia Maria Silva Ataide

Academic Editor

PLOS ONE

**Journal Requirements:**

3. I am very sorry to hear that Harry P. Andreassen has passed away. Could you please confirm if there is any family member or next of kin we should contact if the manuscript is accepted for publication?

**Comments to the Author**

1. Is the manuscript technically sound, and do the data support the conclusions?

Reviewer #1: Yes

Reviewer #2: Yes

2. Has the statistical analysis been performed appropriately and rigorously? 

Reviewer #1: No

Reviewer #2: Yes

3. Have the authors made all data underlying the findings in their manuscript fully available?

Reviewer #1: No

Reviewer #2: Yes

4. Is the manuscript presented in an intelligible fashion and written in standard English?

Reviewer #1: Yes

Reviewer #2: Yes

5. Review Comments to the Author

Reviewer #1: A review of a manuscript entitled “Influence of light availability and soil productivity on insect herbivory on bilberry (Vaccinium myrtillus L.) leaves following mammalian herbivory” for PLoS ONE (PONE-D-19-24380)

General comments

In this study, the authors aimed to evaluate whether the palatability of bilberry leaves (Vaccinium myrtillus) to insect herbivores is influenced by light availability, by soil productivity and by previous herbivory by mammals. A set of generalized linear models were used to test these three variables using a robust sample design. However, the results and discussion are dense, with many figures to look at and re-analyzes being described in the results and discussion. All of this part of data re-analysis to find the best model is part of the methods and I think it should not appear in the results. By reading the methods, the reader expects to see such analyzes and imagines such figures, but during the results the reader finds new analyzes being made based on the previous results. That does not seem appropriate to me here. These should all be described in the methods (or in supplementary documents) and the main results should be presented. This brings me to another issue. besides the text being dense in number of results, there are too many figures (14). In my opinion some tables and figures could be added as supplementary documents (e.g. Table 1, Figures 2, 3, 6, 8, and some ‘re-analyzed’ figures). This ‘cleaning’ of the paper (i.e. reduction in the amount of results by focusing on the main ones) are my main suggestions, since the study deals with an interesting topic and the results are promising. Minor suggestions are outlined below.

Specific comments (L: lines)

L 73-75: I suggest not leaving a sentence “alone” in the text. Please, add it to a paragraph.

L 189-190: What is the criterion used to establish these categories? Distribution of data? There is a much greater amplitude in the intermediate category than in the others.

L 474: Beware, the package itself shows nothing.

I could not access the data via DOI.

Reviewer #2: - The paper claims that bilberry palability by insects following mammalian herbivory is affected by leaf nutrient concentration and that this response is intensified by shade levels.

- The claims are properly placed in the context of previous literature but the readability of the introduction can be improved. Please see comments below.

- The analyses performed support their claims. However, figure 8 regarding the correlation between insect and mammalian herbivory must be improved. The analyses show a significant correlation but for some reason that’s not clear when looking at the figure.

- Information on protocols and analyses seems to be complete.

- The paper could be published provided that the authors improve some aspects (please see comments below).

- Raw data is not included, it would be useful to include it.

- Details of the methodology are sufficient.

- Yes, the manuscript is well organized and written clearly.

Comments

Line 54: There is no connection between the previous paragraph and the paragraph starting in this line. There is a phrase missing introducing this new hypothesis. How are connected the constitutive defense and the optimal defense (OD).

Line 60: Similarly as above, the carbon:nutrient balance (CNB) hypothesis comes out of the blue. Could you please introduce with an explanatory phrase on how this hypothesis is connected with the previous one?

Line 140 to 152: Probably it would be easier for the reader if you integrate this information in the text of the introduction to connect your own hypothesis with the one in the literature.

Line 218: Which are the productivity classes? They have not been introduced so far.

Figure 4a: The scales are so different between insect herbivory and mammalian herbivory that it is difficult to see the trend of the effect of shade on mammalian herbivory. Please separate these two figures. Why in the statistics of figure 4 (lines 303 to 310) there is a report of F and P values of years 2013 and 2014? What about year 2015? Please include missing information.

Why mammals feeding on low soil productivity levels?. Also, there was something going on in 2013: -High mammalian herbivory in quadrats with high shade levels than at quadrats with low shade levels. - there was no difference in mammalian herbivory between the three soil productivity classes. While for the other years a higher mammalian herbivory was registered for low soil productivity. Could you comment on that ?

Figure 6. Please explain briefly in the figure title what you mean with “standardized mammalian herbivory”.

Figure 8. Why an R2 value is not presented in the figure. Please include it. Can you please explain in the figure legend what you mean by “adjusted” R2. The correlation does not seem very strong when the shade variable is taken away…Especially taking into account that the Y axis goes only 35%. It should be 100%.

6. PLOS authors have the option to publish the peer review history of their article (what does this mean?). If published, this will include your full peer review and any attached files.

Reviewer #1: No

Reviewer #2: No

---

## [Author Response · Author response to Decision Letter 0]

26 Jan 2020

Journal Requirements:

Response MSG: I have studied the style templates in detail and have used these style requirements in the manuscript.

Response MSG: I added this sentence: No permits for field site access were necessary, according to Norwegian law (friluftsloven: LOV-1957-06-28-16) that permitted access by foot to natural areas.

3. I am very sorry to hear that Harry P. Andreassen has passed away. Could you please confirm if there is any family member or next of kin we should contact if the manuscript is accepted for publication?

Response MSG: Not necessary to contact any relative of Harry, thanks for the offer. 

Comments to the Author

1. Is the manuscript technically sound, and do the data support the conclusions?

Reviewer #1: Yes

Reviewer #2: Yes

Response MSG: no comment.

2. Has the statistical analysis been performed appropriately and rigorously? 

Reviewer #1: No

Reviewer #2: Yes 

Response MSG: I assume the reason for this 'no' (reviewer #1) is explained under '5. Review Comments to the Author', see my response there.

3. Have the authors made all data underlying the findings in their manuscript fully available?

Reviewer #1: No

Reviewer #2: Yes

Response MSG: All data underlying the findings in the manuscript are fully and freely available via the following URL after the manuscript has been accepted for publication:

https://doi.org/10.18710/89MLBP

Before publication, this information can be accessed via a private URL: 

https://dataverse.no/privateurl.xhtml?token=87b6389f-b5cd-4453-9684-36b5cc1cc3f5

4. Is the manuscript presented in an intelligible fashion and written in standard English?

Reviewer #1: Yes

Reviewer #2: Yes

Response MSG: no comment.

5. Review Comments to the Author

Reviewer #1: A review of a manuscript entitled “Influence of light availability and soil productivity on insect herbivory on bilberry (Vaccinium myrtillus L.) leaves following mammalian herbivory” for PLoS ONE (PONE-D-19-24380)

General comments

In this study, the authors aimed to evaluate whether the palatability of bilberry leaves (Vaccinium myrtillus) to insect herbivores is influenced by light availability, by soil productivity and by previous herbivory by mammals. A set of generalized linear models were used to test these three variables using a robust sample design. However, the results and discussion are dense, with many figures to look at and re-analyzes being described in the results and discussion. All of this part of data re-analysis to find the best model is part of the methods and I think it should not appear in the results. By reading the methods, the reader expects to see such analyzes and imagines such figures, but during the results the reader finds new analyzes being made based on the previous results. That does not seem appropriate to me here. These should all be described in the methods (or in supplementary documents) and the main results should be presented. 

Response MSG: I have moved all the mentioned text parts to the Methods-section or to Supplementary documents, as suggested below by Reviewer #1.

This brings me to another issue. besides the text being dense in number of results, there are too many figures (14). In my opinion some tables and figures could be added as supplementary documents (e.g. Table 1, Figures 2, 3, 6, 8, and some ‘re-analyzed’ figures). 

Response MSG: I have moved several text parts, tables and figures to supplementary documents. To improve readability, I have changed the method, result and discussion section extensively and unneccessary analyses (not necessary for the research question and the predictions) have been removed from the manuscript.

This ‘cleaning’ of the paper (i.e. reduction in the amount of results by focusing on the main ones) are my main suggestions, since the study deals with an interesting topic and the results are promising. 

Response MSG: This 'cleaning' has been done carefully and extensively, see the revised version of the manuscript.

Minor suggestions are outlined below.

Specific comments (L: lines)

L 73-75: I suggest not leaving a sentence “alone” in the text. Please, add it to a paragraph.

Response MSG: This sentence is about the Growth Rate hypothesis. As we do not use this hypothesis in the manuscript, nor refer to it in the manuscript, I have deleted this sentence.

L 189-190: What is the criterion used to establish these categories? Distribution of data? There is a much greater amplitude in the intermediate category than in the others.

Response MSG: We expected the high and low values to have most impact, and in addition they may suffer least from sampling errors, so we made the very low and very high categories narrow. This is now explained in Methods.

L 474: Beware, the package itself shows nothing.

Response MSG: I have changed the sentence 'The DHARMa package showed ...' into 'Analyses with the DHARMa package showed ...'. I have done the same at line 684 (line number as in the first submitted version).

I could not access the data via DOI.

Response MSG: All data underlying the findings in the manuscript are fully and freely available via the following URL after the manuscript has been accepted for publication:

https://doi.org/10.18710/89MLBP

Before publication, this information can be accessed via a private URL: https://dataverse.no/privateurl.xhtml?token=87b6389f-b5cd-4453-9684-36b5cc1cc3f5

Reviewer #2: - The paper claims that bilberry palability by insects following mammalian herbivory is affected by leaf nutrient concentration and that this response is intensified by shade levels.

- The claims are properly placed in the context of previous literature but the readability of the introduction can be improved. Please see comments below.

- The analyses performed support their claims. However, figure 8 regarding the correlation between insect and mammalian herbivory must be improved. The analyses show a significant correlation but for some reason that’s not clear when looking at the figure.

Response MSG: The readability of the whole manuscript has been improved by removing less important information, analyses and figures. Figure 8, where Reviewer #2 refers to, has been moved to the Supplementary documents in our revised manuscript, as suggested by Reviewer #1. The direct correlation between insect and mammalian herbivory, as shown in this figure, is weak, but when soil productivity and shade conditions are taken into account the correlation becomes larger (described in the manuscript).

- Information on protocols and analyses seems to be complete.

- The paper could be published provided that the authors improve some aspects (please see comments below).

- Raw data is not included, it would be useful to include it. 

Response MSG: All the data, including the scripts that we have used to obtain the results in the manuscript, are fully and freely available via the following URL after the manuscript has been accepted for publication:

https://doi.org/10.18710/89MLBP

Before publication, this information can be accessed via a private URL: 

https://dataverse.no/privateurl.xhtml?token=87b6389f-b5cd-4453-9684-36b5cc1cc3f5

- Details of the methodology are sufficient.

- Yes, the manuscript is well organized and written clearly.

Comments

Line 54: There is no connection between the previous paragraph and the paragraph starting in this line. There is a phrase missing introducing this new hypothesis. How are connected the constitutive defense and the optimal defense (OD). 

Line 60: Similarly as above, the carbon:nutrient balance (CNB) hypothesis comes out of the blue. Could you please introduce with an explanatory phrase on how this hypothesis is connected with the previous one?

Response MSG: I added two connecting sentences before introducing the three defense hypotheses to improve readability. 

Line 140 to 152: Probably it would be easier for the reader if you integrate this information in the text of the introduction to connect your own hypothesis with the one in the literature. 

Response MSG: The sentences that I have added (see my response to the comment above) connect the provided theoretical information to our own predictions.

Line 218: Which are the productivity classes? They have not been introduced so far.

Response MSG: The soil productivity classes have been introduced in lines 214-217.

Figure 4a: The scales are so different between insect herbivory and mammalian herbivory that it is difficult to see the trend of the effect of shade on mammalian herbivory. Please separate these two figures.

Response MSG: Figure 4 has been changed (is now Figure 3) and this figure has only one axis now. The figure with mammalian herbivory has been removed.

Why in the statistics of figure 4 (lines 303 to 310) there is a report of F and P values of years 2013 and 2014? What about year 2015? Please include missing information.

Response MSG: In the revised manuscript, we removed the analyses on separate years.

Why mammals feeding on low soil productivity levels?. 

Response MSG: In the revised manuscript, we removed the analyses on mammalian herbivory and soil productivity (not part of our research question nor our predictions and this improves readability of the manuscript).

Also, there was something going on in 2013: -High mammalian herbivory in quadrats with high shade levels than at quadrats with low shade levels. - there was no difference in mammalian herbivory between the three soil productivity classes. While for the other years a higher mammalian herbivory was registered for low soil productivity. Could you comment on that ?

Response MSG: In the revised manuscript, we removed the analyses on mammalian herbivory and shade as well as the analyses on mammalian herbivory and soil productivity (both are not part of our research question nor our predictions and this improves readability of the manuscript).

Figure 6. Please explain briefly in the figure title what you mean with “standardized mammalian herbivory”.

Response MSG: The “standardized mammalian herbivory” is mentioned in the main text (line 223-224). Figure 6 has been deleted in the revised manuscript.

Figure 8. Why an R2 value is not presented in the figure. Please include it. Can you please explain in the figure legend what you mean by “adjusted” R2.

Response MSG: The R2-value is mentioned in the figure legend. This whole figure 8 has been moved to Supplementary documents. We present the adjusted R2 instead of the non-adjusted R2 as the adjusted R2 is dependent on the number of variables in the model and adjusts for sample size.

The correlation does not seem very strong when the shade variable is taken away…Especially taking into account that the Y axis goes only 35%. It should be 100%.

Response MSG: I assume that this comment refers to Figure 9 in the submitted manuscript (becomes Figure 5 in the revised manuscript). Yes the variable shade is very important in the predicted results and this is mentioned in our results (lines 320-325) and our conclusions (lines 427-432). The y-axis is deliberately presented up till 35 % as the predicted values have maximum values around the value 30. By presenting values only up to 35% at the y-axis the lines in the panels are easier to distinguish from each other.

---

## [Decision Letter · Decision Letter 1]

3 Mar 2020

Influence of light availability and soil productivity on insect herbivory on bilberry (Vaccinium myrtillus L.) leaves following mammalian herbivory

PONE-D-19-24380R1

Dear Dr. Schrijvers-Gonlag,

We are pleased to inform you that your manuscript has been judged scientifically suitable for publication and will be formally accepted for publication once it complies with all outstanding technical requirements.

With kind regards,

Livia Maria Silva Ataide

Academic Editor

PLOS ONE

Additional Editor Comments (optional):

Reviewers' comments:

Reviewer's Responses to Questions

**Comments to the Author**

1. If the authors have adequately addressed your comments raised in a previous round of review and you feel that this manuscript is now acceptable for publication, you may indicate that here to bypass the “Comments to the Author” section, enter your conflict of interest statement in the “Confidential to Editor” section, and submit your "Accept" recommendation.

Reviewer #1: All comments have been addressed

Reviewer #2: All comments have been addressed

2. Is the manuscript technically sound, and do the data support the conclusions?

Reviewer #1: Yes

Reviewer #2: Yes

3. Has the statistical analysis been performed appropriately and rigorously? 

Reviewer #1: Yes

Reviewer #2: Yes

4. Have the authors made all data underlying the findings in their manuscript fully available?

Reviewer #1: Yes

Reviewer #2: Yes

5. Is the manuscript presented in an intelligible fashion and written in standard English?

Reviewer #1: Yes

Reviewer #2: Yes

6. Review Comments to the Author

Reviewer #1: Second assessment of the manuscript entitled “Influence of light availability and soil productivity on insect herbivory on bilberry (Vaccinium myrtillus L.) leaves following mammalian herbivory” for Plos One (PONE-D-19-24380R1)

Main comments

In this paper, the authors aimed to assess whether bilberry leaf palatability to insects is affected by light availability, soil productivity, and previous mammalian herbivory. They used a set of generalized linear mixed and additive models to test three main predictions about bilberry leaf palatability using a robust sample design. In my first assessment, I found the results and discussion very dense, with many figures to look at and many re-analyzes being described in both results and discussion. In this new version, I recognize the effort made by the authors to make the text shorter and more objective. I have no new suggestions to make. Therefore, I have a favorable opinion on the publication of this study that deals with an interesting research topic. This paper will be a good piece of work honoring the memory of Professor Harry P. Andreassen.

Reviewer #2: I have already considered that the first version of the manuscript contained an interesting research question, valuable data and a correct statistical analysis. However, there were problems with readability and with the clarity of some of the figures. The new version of the manuscript is much clearer and more enjoyable to read. It was a wise decision to put some of the figures and data as supplementary material. The authors did answer and clarify the questions I had from the previous version.

7. PLOS authors have the option to publish the peer review history of their article (what does this mean?). If published, this will include your full peer review and any attached files.

Reviewer #1: Yes: Pedro Giovâni da Silva

Reviewer #2: Yes: Karen Muñoz Cárdenas

---

## [Editor Report · Acceptance letter]

13 Mar 2020

PONE-D-19-24380R1 

Influence of light availability and soil productivity on insect herbivory on bilberry (*Vaccinium myrtillus* L.) leaves following mammalian herbivory 

Dear Dr. Schrijvers-Gonlag:

I am pleased to inform you that your manuscript has been deemed suitable for publication in PLOS ONE. Congratulations! Your manuscript is now with our production department. 

With kind regards,

on behalf of

Dr. Livia Maria Silva Ataide 

Academic Editor

PLOS ONE